# A New Method for Estimating Atmospheric Turbulence from Global High-Resolution Radiosonde Data and Comparison with the Thorpe Method

Han-Chang Ko<sup>1</sup>, Hye-Yeong Chun<sup>1</sup>

<sup>1</sup> Department of Atmospheric Sciences, Yonsei University, Seoul, South Korea *Correspondence to*: Prof. Hye-Yeong Chun (chunhy@yonsei.ac.kr)

Abstract. This study proposes a new method for estimating atmospheric turbulence from high vertical-resolution radiosonde data (HVRRD) using the minimum Richardson number ( $Ri_{min}$ ). While previous studies using HVRRD have primarily been based on the Thorpe method, which detects turbulence only in regions of local potential temperature overturning (Ri < 0) and does not explicitly account for wind shear, the proposed approach overcomes these limitations. By incorporating the effects of gravity waves on the static stability and vertical wind shear, this method enables the detection of turbulence not only in regions of Ri < 0 but also within statically stable layers characterized by strong shear (0 < Ri < 0.25), where Kelvin-Helmholtz instability is likely to occur. Additionally, comparison with turbulence observations from commercial flights demonstrates that the time series of turbulence derived from the Rimin method exhibits a significantly higher positive correlation with flight observations than that derived from the Thorpe method. Utilizing 10 years of global operational HVRRD, this study further analyzed the climatological distributions of turbulence derived from the Ri<sub>min</sub> method. Results show that turbulence under positive Ri conditions occurs most frequently in winter and less frequently in summer, reflecting the seasonal variability of the jet stream. In contrast, negative Ri cases exhibit a summertime maximum and wintertime minimum in the troposphere, and the opposite seasonal variation in the stratosphere. Regionally, turbulence is most pronounced over Asia, South America, and Antarctica for both positive and negative Ri cases. We upload the datasets produced from the current work at publicly available sites: https://doi.org/10.5281/zenodo.16899801 (Ko and Chun, 2025a), https://doi.org/10.5281/zenodo.16899803 (Ko and Chun, 2025b), https://doi.org/10.5281/zenodo.16899805 (Ko and Chun, 2025c), https://doi.org/10.5281/zenodo.16810246 (Ko and Chun, 2025d), and https://doi.org/10.5281/zenodo.16899789 (Ko and Chun, 2025e).

## 1 Introduction

With the increasing availability of high vertical-resolution radiosonde data (HVRRD) that provide observations of atmospheric quantities at 1- or 2-s intervals (corresponding to vertical resolutions of approximately 5- or 10-m) across the globe, research using such fine-scale vertical resolution are expanding (e.g., Ko et al., 2024). For example, traditional studies using operational radiosondes with relatively coarse resolution (at standard pressure levels and significant levels), such as gravity waves (Ki and Chun, 2010; Zhang et al., 2022), tropopause structure (Sunilkumar et al., 2017), planetary boundary layer height (Guo et al., 2016) and cloud layer estimation (Zhang et al., 2010), are expanded by including atmospheric turbulence research that requires higher vertical resolution of 1- or 2-s (Clayson and Kantha, 2008; Lee et al., 2023; Ko et al., 2024). HVRRD have also been used for numerical weather prediction (NWP) models as one of the main observational datasets for data assimilation (Ingleby et al., 2016) as well as for model validation (Houch et al., 2010).

Atmospheric turbulence is a key process in the exchange of momentum, heat, and energy across different scales of atmospheric motion. Despite its fundamental role, turbulence remains challenging to characterize and predict due to its highly localized, intermittent, and sporadic nature (Clayson and Kantha, 2008; Muhsin et al., 2016; Kohma et al., 2019). Turbulence also poses significant implications for aviation meteorology, as it directly affects aircraft operations and passenger safety. Turbulence cannot be explicitly resolved in current resolution of NWP models and has been parameterized (Janjić, 2001; Shin and Hong, 2011). In operational aviation forecasting, NWP model-derived turbulence diagnostics are used to infer the potential for turbulence (Sharman et al., 2006; Kim et al., 2011; Lee et al., 2020). These approaches inherently depend on the accuracy of the model, making observational validation essential. In particular, global observations of turbulence are crucial for the development and verification of aviation turbulence forecasting systems that integrate multiple diagnostics (Sharman et al., 2006; Sharman and Pearson, 2017; Bechtold et al., 2021; Lee D.-B. et al., 2022; Shin et al., 2023; Ko et al., submitted to JGR: Atmospheres).

The Thorpe method (Thorpe, 1977; 2005) is one of the most widely used methods for estimating turbulence from HVRRD (Ko et al., 2019; Kohma et al., 2019; Zhang et al., 2019; Ko et al., 2024). This method was developed originally in oceanography and identifies turbulent layers based on density overturning in stably stratified fluids, and later, Clayson and Kantha (2008) extended this approach to the atmosphere by applying to the potential temperature profiles. In this framework, the turbulent energy dissipation rate  $\varepsilon$  is estimated as follows:

$$\varepsilon \sim L_T^2 N^3$$
 (1)

where  $L_T$  is the Thorpe length scale, and N is the Brunt–Väisälä frequency of the background atmosphere. Methodological advancements have also been made, such as accounting for the enhancement of  $L_T$  by latent heat release in water-vapor saturated layers (Wilson et al., 2013) and considering instrumental noise on removing false overturning layers (Wilson et al., 2010; 2011).

65

70

90

Recently, Kantha (2024; hereafter K24) questioned the interpretation of  $L_T$ , which has been considered as a length scale in the previous studies (Clayson and Kantha, 2008; Ko et al., 2024), and proposed a reinterpretation of  $L_T$  as a velocity scale. Based on this reinterpretation, he suggested that the turbulence length scale should instead be determined by the background atmospheric stability. While the original equation (1) remains valid in conditions dominated by buoyancy (i.e., strong static stability), alternative formulations are suggested in environments characterized by strong shear or convective instability (K24):

$$\varepsilon \sim L_T^2 N^2 VWS$$
 for shear-driven turbulence, (2)

$$\varepsilon \sim (NL_T)^3/D$$
 for convection-driven turbulence, (3)

where VWS is the vertical wind shear of the horizontal winds and D is the convective layer depth.

Although K24's reinterpretation is reasonable, two key limitations remain: (i) the Thorpe method is applicable only in regions where overturning is detected (i.e., convectively unstable condition), despite turbulence in the real atmosphere also occurring under statically stable conditions, and (ii) applying equations (1)–(3) requires identifying the type of instability responsible for the turbulence by estimating background parameters such as N and VWS using HVRRD. Regarding the first limitation, turbulence can develop under statically stable conditions through wave breaking induced by strong VWS (Palmer et al., 1986). In such cases, the flow stability is characterized by the Richardson number, defined as  $Ri \equiv N^2/VWS^2$ . A negative Ri indicates convective instability, while 0 

## 2 Method

95

100

105

## 2.1 Basic theory

 $Ri_{min}$  refers to the lowest possible value of the Ri when the effects of perturbations due to waves are included. Since a lower value of Ri implies a more unstable flow, identifying its minimum value is essential for detecting turbulence. For a monochromatic sinusoidal wave, the impact of wave on the local static stability and vertical wind shear can be expressed as (Palmer et al., 1986)

$$N_{total}^2 = N^2 \left[ 1 + \left( \frac{N}{II} \delta h \right) \cos \phi \right] \tag{4}$$

$$VWS_{total} = VWS \left[ 1 + Ri^{1/2} \left( \frac{N}{U} \delta h \right) \sin \phi \right]$$
 (5)

where the subscript 'total' denotes the sum of the contributions from the background flow and the perturbation, U is the basic-state wind,  $\delta h$  the amplitude of the displacement of the isentropic surface, and  $\phi$  the wave phase.

One can express the total Ri by dividing equation (4) by the square of (5), as

$$Ri_{total} = \frac{N^2 \left[ 1 + \left( \frac{N}{U} \delta h \right) cos \phi \right]}{VWS^2 \left[ 1 + Ri^{1/2} \left( \frac{N}{U} \delta h \right) sin \phi \right]^2} = Ri \frac{1 + \left( \frac{N}{U} \delta h \right) cos \phi}{\left[ 1 + Ri^{1/2} \left( \frac{N}{U} \delta h \right) sin \phi \right]^2}. \quad (6)$$

For  $Ri_{total}$  to be minimized,  $\cos\phi$  and  $\sin\phi$  should be -1 and +1, respectively. However,  $\cos\phi = -1$  and  $\sin\phi = 1$  cannot be satisfied because  $\cos\phi$  and  $\sin\phi$  are  $\pi/2$  out of phase. Nevertheless, following the approach of Palmer et al. (1986), this study neglected the phase difference between equations (4) and (5) in defining  $Ri_{min}$ , such that,

$$Ri_{total} \ge Ri \frac{1 - \left(\frac{N}{U}\delta h\right)}{\left[1 + Ri^{1/2} \left(\frac{N}{U}\delta h\right)\right]^2} \equiv Ri_{min}.$$
 (7)

Parameterization of  $Ri_{min}$  requires additional calculations (e.g., Palmer et al., 1986 for mountain waves and Chun and Baik, 1998 for convective gravity waves). However, it is reasonable to interpret the measured Ri from HVRRD already being include the effects of wave, and therefore, this study used the Ri calculated by HVRRD as  $Ri_{min}$ .

From  $Ri_{min}$ , the diffusion coefficient K is derived based on the first order Smagorinsky closure (Lilly, 1962; Smagorinsky, 1963; Lane and Sharman, 2008; Kim et al., 2019) as

$$K = (C_s L)^2 |Def| \sqrt{0.25 - \frac{Ri_{min}}{Pr}} \quad (8)$$

where  $C_s$  is set to 0.2 (Lane and Sharman, 2008; Kim et al., 2019), L the length scale, Def the total deformation (Lane and Sharman, 2008), and Pr the Prandtl number specified as 1 following Lane and Sharman (2008) and Kim et al. (2019). Previous modeling studies considered turbulence only when the right-hand side of equation (8) satisfied the condition under  $1 - \frac{Ri_{min}}{Pr} > 0$ , limiting turbulent cases to those with  $Ri_{min} 





In Lilly (1962) and Lane and Sharman (2008), Def is calculated as  $[(\partial u/\partial x - \partial w/\partial z)^2 + (\partial u/\partial z + \partial w/\partial x)^2]^{1/2}$  in two-dimensional x-z plane, where u and w are zonal and vertical wind speed, respectively. However, such calculations are not feasible with HVRRD, as horizontal differences in x-direction are unavailable, and w is not directly observed. To address this limitation, this study performed a scale analysis with  $\partial u \sim 10 \text{ m s}^{-1}$ ,  $\partial x \sim 10^6 \text{ m}$ ,  $\partial w \sim 10^{-2} \text{ m s}^{-1}$ ,  $\partial z \sim 10^4 \text{ m}$ , and found that Def can be approximated by  $\partial u/\partial z$ , the same as VWS in x-z plane. Accordingly, in this study, the total deformation Def is replaced by VWS. Here, the scale analysis was conducted using values for large-scale atmospheric motions (Holton and Hakim, 2013) to reflect the background conditions of large-scale flow responsible for turbulence generation. Lane and Sharman (2008) also notated that  $Ri = N^2/Def^2$ .

During visual inspection, a number of unrealistically high  $\varepsilon$  values were identified, which were likely caused by erroneous vertical temperature gradients (Ko et al., 2024). These gradients can artificially generate potential temperature overturning, leading to spuriously large estimates of  $\varepsilon$ . Given the difficulty in defining a universal threshold for vertical temperature gradients in HVRRD, we excluded extreme values by retaining only turbulence cases with  $\varepsilon$  values smaller than the 99.99<sup>th</sup> percentile (2.183 m<sup>2</sup> s<sup>-3</sup>) of the total cases, following the approach of Ko et al. (2024). Additionally, strong negative Ri values may result in extremely large values of K in equation (8), which leads to unrealistically strong estimates of  $\varepsilon$ . To mitigate this issue, only cases of Ri values greater than or equal to -10 were considered in this study. Cases with Ri less than -10 accounted for approximately 3.5% of the total turbulence events. This lower bound of -10 may be somewhat arbitrary, therefore, additional sensitivity tests were conducted using alternative lower bounds of -100 and -50. Based on visual check, the distribution of  $\varepsilon$  most closely follows a log-normal distribution when the -10 is applied (not shown).

## 3 Data

# 3.1 High vertical-resolution radiosonde data (HVRRD)

This study uses the HVRRD of Ko et al. (2024), although the analysis period extends to 10 years from 2015 to 2024. The
European Centre for Medium-Range Weather Forecasts (ECMWF) has provided global radiosonde observations via the U.S.
National Centers for Environmental Information (NCEI) since October 2014. The data availability period differs across countries and stations; this is illustrated in Fig. S1 of the Supplemental Material in Ko et al. (2024). In this study, only 1-s and 2-s resolution data collected over the 10-year period (2015–2024) from 336 stations were used as HVRRD, and the locations of these stations are shown in Fig. 1. The total number of HVRRD profiles used was 1,346,894.



Figure 1: Locations of radiosonde stations that provided HVRRD used in this study over a 10-year period (2015-2024).

As described in detail in Ko et al. (2024), the radiosonde data used in this study underwent radiation correction and smoothing procedures (Ingleby et al., 2016). Although radiation corrections have been addressed in several studies (e.g., von Rohden et al., 2022; Lee S.-W. et al., 2022), the smoothing algorithms remain proprietary to the instrument manufacturers and are not publicly disclosed (Wang and Geller, 2025). Notably, Wang and Geller (2025) reported that temperature fluctuations in processed data can be reduced by up to a factor of two compared to raw data, with variations depending on the radiosonde instruments.

The HVRRD employed in this study have time resolutions of 1- and 2-s, but the vertical resolution is not uniform due to variations in the ascent rate of the radiosondes. To ensure consistency in vertical spacing, the 1- and 2-s resolution data were interpolated to 5- and 10-m intervals, respectively (Ko et al., 2024).

Instrumental noise can affect the potential temperature, resulting in regions of spurious  $N^2 





motion. Due to the application of a 60-m moving average, all results will be presented only in cases where the thickness of the turbulence layer equals or greater than 60-m.

## 3.2 In-situ flight EDR (flight-EDR) data

As a validation of the newly suggested turbulence estimation in this study, flight-EDR (eddy dissipation rate; EDR =  $\varepsilon^{1/3}$ ) observations obtained from commercial aircraft are used. EDR is a standard aviation turbulence metric and aircraft-independent measure of atmospheric turbulence (ICAO, 2010; Sharman et al., 2014). The flight-EDR data are derived using a vertical wind-based turbulence estimation algorithm, as aircraft are more sensitive to vertical gusts than horizontal ones (Hoblit, 1988; Sharman et al., 2014). Several types of reporting strategies are employed: 15-min routine reports, which are reported at regular intervals, and "triggered" reports, which are reported when the estimated EDR exceeds a preset threshold (0.18 m<sup>2/3</sup> s<sup>-1</sup>). Additionally, some aircrafts generate 1-min routine reports, although these may include interpolated null values. However, the dataset used here does not indicate whether a report was interpolated or not. Therefore, all available reports are used without filtering. For consistency with the HVRRD-derived turbulence—which are only available when turbulence is present—we retain only those flight-EDR reports with EDR values greater than 0 m<sup>2/3</sup> s<sup>-1</sup>. Further details regarding the turbulence estimation algorithm and reporting protocols can be found in Sharman et al. (2014).

# 4 Validation of $Ri_{min}$ -based turbulence estimated from HVRRD

## 4.1 A case study

To evaluate the validity of the proposed method, a case study was first examined in Fig. 2. As shown in Fig. 2g, the  $Ri_{min}$  method identifies all three turbulent layers with Ri 

Figure 2: Vertical profiles observed at Grand Junction station (39.12°N, 108.53°W), CO, USA, at 12 UTC on April 19, 2023: (a) potential temperature, (b) zonal and meridional winds, (c) squared Brunt-Väisälä frequency, (d) vertical wind shear, (e) Richardson number, (f) Thorpe-estimated  $\varepsilon$ , and (g)  $Ri_{min}$ -based  $\varepsilon$ . In (f), green, blue, and red represent  $\varepsilon$  estimated using the conventional Thorpe method, the shear-driven assumption, and the convection-driven assumption based on Kantha (2024), respectively. In (g), red and blue indicate  $\varepsilon$  for negative and positive Ri cases, respectively. The horizontal dashed line in each plot denotes the tropopause height.

### 4.2 Comparison with flight-EDR data


As an additional validation, this study examined the monthly time series of the ratio of moderate-or-greater (MOG) intensity turbulence events, defined as EDR  $\geq 0.22~{\rm m}^{2/3}~{\rm s}^{-1}$  (Sharman and Pearson, 2017). Following Ko et al. (2023), who compared flight-EDR reports with HVRRD-EDR estimated using the Thorpe method over major U.S. routes, this study applied the same framework using the  $Ri_{min}$ -based method, alongside the Thorpe method for comparison. Specifically, major flight routes with dense flight-EDR reports were identified, then HVRRD-EDR were computed from HVRRD profiles collected at radiosonde stations located within those main flight routes (see Fig. S1 in the Supplementary material). Considering that HVRRD are available at 00 and 12 UTC, only flight-EDR reports observed within  $\pm 1$  hour of those times were used.




Figure 3 compares the monthly MOG ratios derived from flight-EDR and those of HVRRD-EDR derived from the  $Ri_{min}$  method and the Thorpe methods. The flight-EDR shows seasonal peaks (up to 2.5%) in late winter and early spring, likely linked to enhanced jet activity and wind shear (Sharman et al., 2014). The  $Ri_{min}$  method captures this seasonality well, showing a strong and statistically significant correlation with flight-EDR (r = 0.76). In contrast, the conventional Thorpe method shows almost no correlation (r = -0.03), indicating poor performance in representing MOG variability revealed in aircraft measurements. The Thorpe method under the shear-driven assumption (K24-shear; green) improves the correlation (r = 0.38), though still underperforms relative to the  $Ri_{min}$  method. The Thorpe method under the convection-driven assumption (K24-conv; orange) persistently underestimates MOG ratios (below 0.5%) and shows a negative, insignificant correlation (r = -0.15), likely due to its mismatch with the clear-air turbulence (CAT) nature of flight-EDR observations (Sharman et al., 2014; Ko et al., 2023). These results indicate that the  $Ri_{min}$  method more effectively captures the climatological variability of MOG intensity turbulence than the Thorpe method.

Figure 3: Time series of the monthly moderate-or-greater (MOG) intensity ratio for flight-EDR (black),  $Ri_{min}$ -EDR (red), and EDR estimated using the conventional Thorpe method (blue), the shear-driven assumption (green), and the convection-driven assumption (orange) based on Kantha (2024). The data cover the period from August 2017 to December 2024 (89 months) at z = 20—40 kft within the main flight routes in the United States. "r" denotes the correlation coefficient between the MOG ratios of each EDR and flight-EDR, with \* indicating statistical significance at the 99% confidence level.




## 5 Global distributions of the $Ri_{min}$ -based turbulence

This study further investigated the global distributions of  $Ri_{min}$ -based turbulence using HVRRD (2015–2024). Figure 4 presents seasonal occurrence frequency distributions of  $log_{10}\varepsilon$ , defined as the number of turbulence events per bin normalized by the total number of HVRRD profiles in each domain. The panels distinguish between the Northern Hemisphere (NH) and Southern Hemisphere (SH), and between the troposphere (surface to tropopause) and stratosphere (tropopause to 30 km). Tropopause height was calculated for each profile using the WMO (1957) definition of the first tropopause. For each panel,  $log_{10}\varepsilon$  distributions are shown separately for positive and negative Ri cases.

Across all domains,  $\log_{10}\varepsilon$  ranges from -6 to -1 m<sup>2</sup> s<sup>-3</sup>, with peaks for positive Ri cases near -4. Negative Ri cases are slightly right-shifted, particularly in the troposphere, indicating that stronger turbulence is likely associated with convective overturning. In the stratosphere, negative Ri cases are less frequent but reach higher  $\log_{10}\varepsilon$  values than positive Ri cases, suggesting that while overturning is less likely to occur than non-overturning turbulence in the statically stable stratosphere, the intensity of turbulence that overcomes such strong stability and develops into overturning is greater than that of turbulence in positive Ri cases. The global mean of  $\log_{10}\varepsilon$  is -3.37 m<sup>2</sup> s<sup>-3</sup>, (-3.46 for positive Ri; -3.35 for negative Ri), consistent with previous studies based on the Thorpe method (Clayson and Kantha, 2008; Ko et al., 2024), aircraft (Sharman et al., 2014), and radar (Li et al., 2016) observations.

The positive *Ri* regime exhibits a clear seasonal cycle, with higher occurrences in winter (DJF in the NH, JJA in the SH) and lower in summer (JJA in the NH, DJF in the SH), reflecting enhanced shear-driven turbulence due to strong upper tropospheric and lower stratospheric (UTLS) jets (Koch et al., 2006). In the troposphere (Figs. 4a, 4c), negative *Ri* cases are most frequent during summer and least frequent in winter in both hemispheres, supporting the interpretation that turbulence in this regime is predominantly convectively generated. In contrast, stratospheric negative *Ri* cases (Figs. 4b, 4d) occur most frequently in winter in both hemispheres, suggesting that shear instability plays a dominant role in driving overturning turbulence in the stratosphere.



Figure 4: Occurrence frequencies of  $\log_{10}\varepsilon$  in (a) the Northern Hemisphere (NH) troposphere, (b) the NH stratosphere, (c) the Southern Hemisphere (SH) troposphere, and (d) the SH stratosphere over a 10-year period (2015–2024). In each plot, the left and right sides represent the results for positive Ri (PosRi) and negative Ri (NegRi) cases, respectively. The black, green, red, and blue lines indicate the results for December–January–February (DJF), March–April–May (MAM), June–July–August (JJA), and September–October–November (SON), respectively. "n" denotes the total occurrence number of  $\log_{10}\varepsilon$  in each domain.

Figure 5 presents the occurrence frequency distributions of L, which characterizes the depth of mixing for momentum and trace gases and is a fundamental variable in turbulence modeling (Dewan, 1981; Osman et al., 2016; Muñoz-Esparza et al., 2020; Ko et al., 2024). Across all domains, the occurrence frequency of L exhibits an approximately exponential decay with increasing L. In the troposphere (Figs. 5a, 5c), negative Ri cases reach thicker layers (up to 1,000 m) than positive Ri cases (up to 600 m), suggesting convectively generated turbulence extends deeper vertically than shear-driven turbulence. In the stratosphere (Figs. 5b, 5d), this contrast remains but is less pronounced, likely due to the stronger static stability that suppresses deep convective overturning.

Globally, the  $50^{th}$  (median) and  $95^{th}$  percentiles of L are 90 and 225 m, respectively. These values are smaller than those reported by Ko et al. (2024). For positive Ri cases, the  $50^{th}$  and  $95^{th}$  percentiles are 80 m and 180 m; for negative Ri cases, they are 100 m and 270 m, reinforcing that overturning turbulence tends to produce thicker layers than those of positive Ri



conditions. Seasonal variations in L are less prominent than those of  $\log_{10}\varepsilon$  (Fig. 4), so each panel includes a zoomed-in view up to the global  $50^{th}$  percentile (90 m) to better examine seasonal differences. The seasonal variations in L are consistent with those observed for  $\log_{10}\varepsilon$ . In the troposphere, positive Ri cases exhibit higher occurrence in winter, likely due to stronger jet-induced wind shear. A more quantitative assessment of the relationship between jet strength and turbulence occurrence frequency is provided later in Fig. 6. For negative Ri, a rightward shift in summer indicates deeper turbulent layers, consistent with enhanced convective activity during warmer months.

Figure 5: The same as in Fig. 4, but for the thickness of the turbulence layer L.

To further examine the vertical distribution of turbulence and its relationship with wind speed, Fig. 6 presents vertical profiles of wind speed and turbulence occurrence frequency. The vertical axis is aligned relative to the altitude of maximum wind speed ( $Z_{\text{WSPDmax}}$ ) to account for variations in jet core height across different atmospheric conditions and latitudes. Here, the jet strength and  $Z_{\text{WSPDmax}}$  are determined following Koch et al. (2006): the horizontal wind speed (WSPD) averaged over two pressure levels is defined by  $\alpha vel \equiv \frac{1}{p_2 - p_1} \int_{p_1}^{p_2} (u^2 + v^2)^{1/2} dp$ , where u and v are the zonal and meridional wind,




respectively, and  $p_1$  and  $p_2$  are set to 400 and 100 hPa, respectively. Within this 400–100 hPa, the altitude at which the maximum WSPD is defined as  $Z_{WSPDmax}$ . Then,  $\alpha vel \geq 30 \text{ m s}^{-1}$  and  $\alpha vel 

Figure 6: Vertical profiles of (a) horizontal wind speed and (b, c) nonzero turbulence frequency for positive and negative *Ri* cases, respectively, relative to the altitude of maximum wind speed (Z<sub>WSPDmax</sub>) over a 10-year period (2015–2024). Blue and red lines represent the results for weak and strong jet-streams, respectively, as defined in the main text. Solid and dashed lines correspond to winter (DJF in the NH and JJA in the SH) and summer (JJA in the NH and DJF in the SH), respectively.

Previous studies (Ko and Chun, 2022; Ko et al., 2024) have shown that a simple arithmetic mean of  $\log_{10}\varepsilon$  does not properly represent turbulence characteristics, particularly in the stratosphere where turbulence is infrequent and a few strong  $\varepsilon$  events






can dominate the average. In contrast, the troposphere generally exhibits greater turbulence occurrence and thicker turbulence layers (see Fig. 5). To better characterize layer-averaged turbulence, this study adopts the effective  $\varepsilon$  (EE) introduced by Ko and Chun (2022), which integrates both the turbulence intensity ( $\varepsilon$ ) and the vertical extent (L) of each turbulent layer. EE reflects the atmospheric portion affected by turbulence within a given altitude range, and is defined as EE =  $\sum (\varepsilon \times L)/Z$  in units of [m<sup>2</sup> s<sup>-3</sup>], where Z is the depth of the layer considered. In Fig. 7, Z is approximately 11.4 km in the troposphere and 18.3 km in the stratosphere, while in Fig. 8, Z is uniformly 1 km.

Figure 7 shows the global distribution of  $log_{10}EE$  at individual radiosonde stations. The highest values appear in negative Ri cases within the troposphere (Fig. 7c), while the lowest values occur in the stratosphere for negative Ri regime (Fig. 7d). Positive Ri cases (Figs. 7a, 7b) yield similar  $log_{10}EE$  values in both layers. These patterns indicate that turbulence associated with convective instability in the troposphere is stronger than that generated by shear-driven processes. In Fig. 7a, the distribution of  $log_{10}EE$  for positive Ri cases in the troposphere shows intermediate values across most regions, with relatively higher values over Türkiye, the eastern U.S., Central Europe, East Asia, and the southern Andes. Lower values appear in the tropics and northern high latitudes. The stratospheric counterpart (Fig. 7b) exhibits a similar spatial pattern but with even lower at high latitudes of both hemispheres.

In Fig. 7c, negative Ri cases in the troposphere show that  $\log_{10}EE$  values are strong over Türkiye, the western United States, East Asia, and South America. These regions have significant and complex terrain, suggesting that turbulence could originate from orographic sources such as orographically induced convection and the breaking of convective and mountain gravity waves. In contrast, stations in the tropics and polar regions show relatively lower values compared to the other regions. In Fig. 7d, the lowest  $\log_{10}EE$  values are revealed among the four domains, reflecting the rarity of negative Ri cases in the highly stratified stratosphere. Slightly elevated values are seen over Türkiye, East Asia, and South America, while lower values are observed over the eastern U.S., Northern Europe, Australia, and Antarctica. The spatial pattern of  $\log_{10}EE$  with negative Ri cases is consistent with that from the Thorpe method in Ko et al. (2024) in terms of the range of values, the maximum over Türkiye, and locally large values over East Asia and South America. This agreement likely stems from the conceptual similarity between both methods, which identify turbulence under statically unstable conditions (i.e.,  $N^2 

Figure 7: Global distributions of  $log_{10}EE$  at each station over a 10-year period (2015–2024) for (a) positive Ri (PosRi) cases in the troposphere, (b) PosRi cases in the stratosphere, (c) negative Ri (NegRi) cases in the troposphere, and (d) NegRi cases in the stratosphere.

Figure 8 presents the vertical distribution of  $log_{10}EE$  in 1-km bins relative to the tropopause height  $z - z_{tp}$  to consider the latitudinal and seasonal variations in the tropopause (Birner, 2006). Here, z is the geometric altitude above mean sea level and  $z_{tp}$  is the tropopause height of each HVRRD profile.

For positive Ri cases, the global median  $log_{10}EE$  exhibits a smooth profile with little contrast between the troposphere and stratosphere. Peak median values occur near the tropopause, indicating enhanced turbulence activity in this UTLS. In contrast, negative Ri cases show a distinct structure:  $log_{10}EE$  peaks in the lower troposphere, where convective instability is more likely to occur due to surface heating and buoyancy, then decreases sharply with altitude. Comparing the two regimes of Ri, median  $log_{10}EE$  is higher for negative Ri cases in the troposphere, whereas positive Ri cases dominate in the stratosphere—highlighting different turbulence generation mechanisms in the two regimes. Moreover, negative Ri cases exhibit a broader spread in  $log_{10}EE$  across most altitudes, reflecting the more variable and intermittent nature of overturning turbulence. The

vertical structure for negative *Ri* cases is consistent with the Thorpe method results in Ko et al. (2024), as in the horizontal distributions in Fig. 7.

Figure 8: Vertical distributions of  $log_{10}EE$  within each 1-km altitude bin relative to the tropopause over a 10-year period (2015–2024). Thick lines represent the median values of  $log_{10}EE$ , while the hatched areas indicate the full range. Blue and red represent the results for positive and negative Ri cases, respectively.





To assess regional variability, Fig. 9 presents statistics of log<sub>10</sub>EE across 10 globally defined regions (Fig. 9a), determined based on geographical proximity and non-overlapping station coverage. Figures 9b and 9c show boxplots of log<sub>10</sub>EE for positive and negative *Ri* cases, respectively.

For positive *Ri* cases (Fig. 9b), regional differences are relatively small both in the troposphere and the stratosphere. Tropospheric median values are uniform, with slightly elevated values in Regions 5 (Asia), 7 (South America), and 10 (Antarctica). Stratospheric medians are comparable with tropospheric medians, but show greater inter-regional variability as represented by wider dashed lines in the stratosphere than in the troposphere. A similar pattern is observed for negative *Ri* cases (Fig. 9c), consistent with Ko et al. (2024), likely due to broader horizontal drifting of radiosondes at higher altitudes. Tropospheric EE values for negative *Ri* are notably higher than those for positive *Ri*, again peaking in Regions 5, 7, and 10, reaffirming that turbulence is generally more vigorous and variable under unstable stratification than under stable conditions. In the stratosphere, by contrast, negative *Ri* cases exhibit considerably lower EE values than in the troposphere, primarily due to their rarity in the stratosphere.

Interestingly, both positive and negative *Ri* cases in the troposphere show the global maximum of median log<sub>10</sub>EE in Region 10 (Antarctica), contrasting with the stratospheric minimum over the same region. This also differs from Ko et al. (2024), where Antarctica showed the lowest EE in the troposphere. This discrepancy likely stems from different definitions of the troposphere: the present study includes all levels up to the tropopause, while Ko et al. (2024) exclusively considered in the free troposphere (3 km above the station to the tropopause). When our calculations are restricted to the free troposphere (Fig. S2 in the Supplementary material), Antarctic values decreased substantially, aligning with Ko et al. (2024). This implies that EE values in the lower troposphere over Antarctica are particularly large, and these elevated values may reflect strong orographic effects. Yoo et al. (2020) showed that steep slopes near the Antarctic edge enhance frontogenesis, particularly at 850 hPa. Since most Antarctic HVRRD stations are located along these margins (Fig. 1), enhanced turbulence in this region may be influenced by orographic effects and associated dynamical processes. Additionally, under the free troposphere definition, the highest EE appears over Region 5 (Asia), consistent with Ko et al. (2024).

Figure 9: (a) Division of radiosonde stations into Regions 1–10, with the number of stations and profiles in each region over a 10-year period (2015–2024). (b, c) Median (dots) and full range (dashed lines) of log<sub>10</sub>EE in each region in the troposphere (blue) and stratosphere (red), shown separately for positive *Ri* (PosRi) and negative *Ri* (NegRi) cases.

## 6 Data availability

The turbulence data obtained from the current study based on the  $Ri_{min}$  method using high vertical-resolution radiosonde data (HVRRD) for 2015–2024 are available at https://doi.org/10.5281/zenodo.16899801 (Ko and Chun, 2025a), https://doi.org/10.5281/zenodo.16899803 (Ko and Chun, 2025b), https://doi.org/10.5281/zenodo.16899805 (Ko and Chun, 2025c), https://doi.org/10.5281/zenodo.16899806 (Ko and Chun, 2025c), https://doi.org/10.5281/zenodo.16899789 (Ko and Chun, 2025e). The HVRRD used in this study to estimate turbulence are openly available from the National Centers for Environmental Information (NCEI, 2025; https://www.ncei.noaa.gov/data/ecmwf-global-upper-air-bufr/). In-situ flight EDR data are available from the National Oceanic and Atmospheric Administration (NOAA)'s Meteorological Assimilation Data Ingest System (MADIS) site (NOAA MADIS, 2025; https://madis-data.cprk.ncep.noaa.gov/madisPublic1/data/archive/).

## 7 Summary and conclusions





By utilizing HVRRD to estimate atmospheric turbulence, this study introduced a new method based on  $Ri_{min}$ , addressing key limitations of the widely used Thorpe method. While the Thorpe method identifies turbulence associated with static instability (i.e.,  $N^2 



Science Solution Discussions Data

these observations are concentrated along major flight routes and tend to avoid regions of strong convective activity, resulting in spatial sampling biases (Ko et al., 2023). Under this situation, turbulence estimated from HVRRD, such as from the current study, offers a complementary observational dataset, especially in regions where aircraft observations are limited or unavailable. Ultimately, this application has the potential to contribute to safer and more efficient flight operations.

Importantly, the present study used publicly available high-resolution radiosonde data provided by operational stations. As more radiosonde stations offer 1- or 2-second resolution data, future research on atmospheric turbulence will benefit from increased spatial coverage. It is highly imperative that high-resolution data be made available for international use, although some countries still do not provide HVRRD internationally (Ko et al., 2024). Broader data sharing of HVRRD will greatly benefit not only turbulence research but also studies on general atmospheric research, including gravity waves, boundary layer structure, tropopause dynamics, and numerical model validation.

### **Author contribution**

HCK performed conceptualization, data curation, formal analysis, investigation, visualization, and writing. HYC performed conceptualization, formal analysis, funding acquisition, investigation, supervision, and writing.

# **Competing interests**

The authors have no competing interests to declare.

## Acknowledgements

This research was supported by the Korea Meteorological Administration (KMA) Research and Development Program under Grant RS-2022-KM220410.

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
