# Peer review of "A New Method for Estimating Atmospheric Turbulence from Global High-Resolution Radiosonde Data and Comparison with the Thorpe Method"

_Earth System Science Data, 2025_

## Referee Comment (RC1)

Review comments to the manuscript, entitled "A New Method for Estimating Atmospheric Turbulence from Global High-Resolution Radiosonde Data and Comparison with the Thorpe Method" (essd-2025-485).

**< Overview >**

This paper has a potential of significant and timely contribution that effectively leverages high vertical-resolution radiosonde data (HVRRD) to create a unique global turbulence dataset. The proposed Ri_min method correctly addresses the limitations of the Thorpe method by including turbulence detection in statically stable, strong shear layers (0 < Ri < 0.25). The paper is well-written and the core methodology is sound, satisfying the main scientific requirements for publication in *Earth System Science Data* (ESSD). I recommend a **Minor Revision** to address justification of practical considerations, Illustrative case study near jet stream, data usability, and ensuring the dataset meets the high standards of long-term data archiving.

**< Minor Comments >**

Here are five specific comments that require minor revisions:

**1. Justification of Two Practical Considerations**

While the methodology for deriving EDR from Ri_min is detailed, two practical considerations of using L and VWS instead of Def need to be justified more clearly.

- The authors mentioned that the Thorpe method has a limitation to explicitly consider convectively unstable condition (Ri < 0), which could have a relatively large mixing length (convective overturning of wave or turbulent eddy). But, shear driven KHI is very intermittent, which would have a small mixing length (small-scale eddy). So, it might be necessary to justify more specifically why the authors adapted the length scale from Thorpe method here?

- The authors substitute the def in Eq. (8) by VWS based on simple synoptic-scale analysis, which would make sense in some part. But, it would not be applicable in a case where strong VWS in anticyclonic shear and curvature jet stream (e.g., Knox 1997). Given the simple assumption that the cyclonic and anticyclonic jet streams are equally happening in mid-latitude, anticyclonic curvature jet stream is theoretically stronger than cyclonic jet based on the gradient wind balance. I think the authors need to more carefully justify (or provide some limitations of current method) using VWS instead of DEF in this study.

**2. Illustrative Case Study for 0 < Ri < 0.25 Turbulence vs in situ EDR near jet stream**

The core scientific advancement of this work is the ability of the Ri_min method to detect turbulence in the statically stable, high-shear region (0 < Ri < 0.25) where the Thorpe method fails. This claim, while theoretically sound, needs compelling direct comparison of Ri_min method with in situ EDR obs. I know it is difficult to find a case

that has collocated pairs of HVSSD and in situ EDR. Considering that jet stream is synoptic scale, it will be great to find a CAT outbreak day (last 1-2 days) near jet stream to be captured by both HVRRD and in situ EDR. Then, it will be more convincing that this new method really show a good performance for KHI near upper-level jet system.

**3. Explicit Declaration of Data Output Format and Internal Structure**

For publication in ESSD, data usability is paramount. The manuscript must explicitly declare the chosen long-term archiving data format and its internal structure for user accessibility. Please state the definitive file format (e.g., NetCDF, HDF5, or another community standard) that will be used for the final dataset. More importantly, provide a clear, dedicated table listing all variable names (e.g., turbulence_edr), their units (e.g., $m^{2/3} s^{-1}$), and the corresponding CF-compliant metadata for each variable provided in the data files.

**4. Clearer Statement on Spatio-Temporal Data Heterogeneity**

The dataset is unique because it is global, but its temporal and horizontal resolution is fundamentally constrained by the heterogeneous global radiosonde network (typically 00/12 UTC, geographically sparse). The current description emphasizes the high vertical resolution but downplays the horizontal and temporal sparseness of the overall product. A single sentence or footnote in the Abstract or Data Description section must clearly state that the dataset's spatial and temporal coverage reflects the limitations and heterogeneity of the operational global radiosonde observing network.

**5. Final Archival Repository and Versioning Commitment**

ESSD requires a strong commitment to long-term data preservation and traceability. Please specify the exact permanent digital repository (e.g., a recognized data center like NOAA NCEI, EUMETSAT, or a major institutional repository with DOIs) where the final dataset will be lodged. Furthermore, provide a clear statement regarding the versioning scheme (e.g., "The dataset presented here will be designated Version 1.0 (v1.0). All future updates will be released as incremental versions, v1.1, v2.0, etc., with associated Digital Object Identifiers (DOIs).").